# Sustainable Recovery of Valuable Nanoporous Materials from High-Chlorine MSWI Fly Ash by Ultrasound with Organic Acids

**DOI:** 10.3390/molecules27072289

**Published:** 2022-03-31

**Authors:** Tam Thanh Nguyen, Cheng-Kuo Tsai, Jao-Jia Horng

**Affiliations:** 1Faculty of Environment, University of Science (VNUHCM), Ho Chi Minh City 700000, Vietnam; 2Vietnam National University Ho Chi Minh City, Ho Chi Minh City 700000, Vietnam; 3Department of Safety, Health, and Environmental Engineering, National Yunlin University of Science and Technology, Yunlin 64002, Taiwan; cktsai@yuntech.edu.tw

**Keywords:** MSWI fly ash, ultrasound, organic acid, dechlorination

## Abstract

The new technology development for municipal solid waste incineration fly ash treatment and reuse is urgent due to landfill shortage and environmental effect of leached hazardous substances. Chlorine (Cl) is worth considering due to its high levels in fly ash. In this study, a treatment process of ultrasound combined with organic acid was used to eliminate Cl from fly ash to enhance its properties for reuse. Taguchi methodology was implemented to design the experiments by controlling four impact factors and the contribution of each factor was evaluated by the ANOVA analysis of variance. Following two treatment steps within 5 min with a solid/liquid ratio of 1:10 at 165 kHz, 98.8% of Cl was eliminated. Solid/liquid ratio was the most prominent factor that contributed to the Cl removal with more than 90%, according to the ANOVA analysis of variance. *Tert*-butyl alcohol (tBuOH), an ^•^OH radical scavenger, was utilized to examine different effects of ultrasonic cavitation on Cl removal efficiency. A 20 kHz ultrasound was used to explore the influence of multi-frequency ultrasound with different mechanical and sonochemical effects on the fly ash dechlorination. This ultrasonic-assisted organic acid treatment was found to be a time and cost-effective pathway for fly ash Cl removal.

## 1. Introduction

Municipal solid waste (MSW) is rapidly expanding into a global environmental issue. In Taiwan, managing MSW is a difficult challenge due to a variety of circumstances, such as limited landfill sites, population density, and strict environmental protection regulations. In addition, municipal solid waste incineration (MSWI) is widely used. According to the statistical report in 2020 from the Environmental Protection Administration of Taiwan, 25 MSWI were in operation and incinerated 3.789 million tons of solid waste, accounting for 39% of total treated solid waste. Through the incineration treatment, up to 90% of the volume of fly ash can be reduced, and viruses and bacteria can be killed at high temperature [1,2]. The amount of hazardous-substance-containing ashes from MSWI was 1.185 million tons per year, which was usually sent to landfills for final disposal. However, in recent years, the recycling and reuse of MSWI fly ash have attracted considerable attention due to the shortage of landfill space and the danger of toxic chemicals that are leached from fly ash into the environment. MSWI fly ash can be recycled for multiple purposes, including the manufacturing of building materials [3], adsorption materials [4,5], zeolite [6,7,8], and hybrid composite [9], as well as serving as seed crystals in the fluidized bed crystallization process for environmental remediation. However, the recycling and reuse of MSWI fly ash are challenged since it is classified as hazardous waste due to the high content of dioxins, soluble metal salts, toxic metals, and alkali chlorides [10] that will harm the environment and human health. In Taiwan, plastic debris and food residue are the major constituents of MSWI, resulting in high-chlorine (Cl) concentration in waste and fly ash. Specifically, in this study, the Cl content of MSWI fly ash is 32.15%. The high quantities of Cl are dangerous, not only during the recycling process, but also when using fly ash-based products. Therefore, a suitable treatment process for Cl elimination from MSWI fly ash to meet the standards is recommended prior to reuse or recycling. However, whereas previous studies have been intensively focused on heavy metals elimination from fly ash, there is still insufficient information regarding MSWI fly ash Cl removal.

Several methods, including washing [11,12,13], leaching [14,15,16], and thermal treatment [17,18,19] have been studied for the removal of Cl from MSWI fly ash to subsequently reuse the fly ash. It has been reported that the water-washing procedure has been recognized as the simplest and most practical technology for Cl removal [20]. The results showed that following two washing steps of 2 h and a single rinse in freshwater, 72% of Cl was eliminated. Moreover, Gau et al. used a four-step water leaching procedure with 5 min for each step to remove Cl from MSWI fly ash [13]. The Cl content could be eliminated by 20–80%. The water-washing process can effectively remove water-soluble Cl. However, water-insoluble Cl salts are difficult to release. The leaching process for the high removal efficiency by acid solvents [15,16] have been investigated. The acid-leaching procedure enables the conversion of water-insoluble Cl into water-soluble Cl, resulting from the fly ash internal structure damage by a high concentration of H^+^. Thereafter, it is released into the solution [21].

Ultrasonic-assisted technology, which is a green and safe technique, has garnered considerable attention recently in the field of wastewater treatment. The collapse of cavitation bubbles caused by ultrasonic irradiation creates localized conditions of severe pressure and temperature, as well as simultaneously facilitates the formation of reactive free radicals and a variety of mechanical effects [22,23]. This indicates that ultrasonic cavitation is advantageous for the degradation of contaminants and wastewater treatment applications, which is attributed to the mechanical disturbance effect and sonochemical effect of ultrasound [24,25,26,27]. It is well recognized that compared with conventional methods, ultrasonic treatment proves the advantages of great efficiency, short reaction time, and nontoxicity since water is used as a solvent [28]. In recent years, the use of ultrasound in the heavy metal leaching process from various kinds of ashes has been reported [29]. The mechanochemical force of ultrasound has shown the prominence to the metal leaching efficiency. Ultrasonic-assisted nitric acid has been applied in the leaching of heavy metals from fly ash and showed 98% Pb, 86% Cd, 73% Cu, and 42% Zn removal with a nitric acid concentration of 5.3 mol/L, fly ash/nitric acid ratio of 1:20, and ultrasonic irradiation time of 4 h [16]. In addition, sulfuric acid exhibited the potential in removing 88% Cd, 51% Cu, and 17% Zn at 4.7 mol/L sulfuric acid, fly ash/sulfuric acid ratio of 1:40, and ultrasonic irradiation time of 4 h [15]. The ultrasound method revealed superior in anions and cations leaching compared with the shacking approach using pure water [30]. It was reported that it required 6 h to obtain 1.99 mg-Cl/100 g-fly ash by shacking-assisted-water leaching, while the extraction time was 15 min by the ultrasound method to achieve that value. The ultrasonic extraction process was also confirmed as independent of the extraction time since the extracted Cl amount was 2.17 and 2.08 mg-Cl/100 g-fly ash after 30 and 45 min irradiation, respectively. In other words, ultrasonic irradiation can quickly remove Cl from fly ash, resulting in the lower time and cost requirement.

The Taguchi methodology is regarded as a statistical approach to optimize the combination of process parameters, improve the quality of material, and reduce the cost for their application in manufacturing [31]. By adopting an orthogonal array, the number of experiments and analysis procedures to find optimal experimental conditions can be reduced, resulting in time and cost savings. In our previous study, Taguchi experiments have been conducted to investigate optimum parameters for ensuring the quality of the material produced from glass fiber wastes and improving the removal efficiency of MB dye [32]. To recover valuable nanoporous materials from high-chlorine MSWI fly ash, there are various parameters and their interactions will influence the process efficiency. It is challenging to examine the effect of all those variables and interactions on the quality of the final product. A large number of experiments and evaluating processes are required to fully determine the best processing parameters. Therefore, Taguchi optimization could be used to design the optimal experimental procedure for creating new valuable fly ash- based nanoporous materials with the highest recovery efficiency, fewer experiments, better material quality, and shorter operating time.

This study aimed to investigate the synergistic effect of ultrasound and acid on the Cl removal rate from MSWI fly ash due to the mechanical and chemical effects of ultrasonic cavitation, which enhance the formation of the pores, increase the surface area, create porous fly ash structure, and simultaneously facilitate the effect of acid on fly ash, thereby enabling effective Cl elimination and consequently obtaining a well-structured and free-toxic material for environmental applications. This as-treated material from MSWI fly ash is considered as a valuable nanoporous material for various applications in construction and environmental engineering. The low-cost organic acids were utilized, including citric acid, ascorbic acid, and ethylenediaminetetraacetic acid disodium salt dihydrate (EDTA-Na_2_). The Cl removal process involves two steps of organic acid leaching and water rinsing assisted by ultrasound. This two-step treatment approach is hereby referred to as the ultrasound-acid-water (UAW) process. The physical and chemical structures of raw and treated MSWI fly ashes were investigated to understand the Cl removal mechanisms by field-emission scanning electron microscopy (FE-SEM), energy dispersive X-ray (EDS), X-ray diffraction (XRD), and Brunauer–Emmett–Teller (BET) surface area analysis. The Taguchi methodology was used to find the optimal experimental conditions for Cl removal with four controllable factors, including organic acid concentration (factor A), ultrasonic irradiation time (factor B), solid/liquid (S/L) ratio (factor C), and vertical position of fly ash sample located in the ultrasonic bath (factor D) at 165 kHz and 600 W. ANOVA analysis of variance was deployed to evaluate the impact of each controllable factor on the Cl elimination rate. For a better understanding of the mechanical and chemical effects of ultrasonic cavitation on the dechlorination efficiency from MSWI fly ash, *tert*-butyl alcohol (tBuOH) was utilized in the experiment as an ^•^OH radical scavenger. Aluminum foil erosion was performed to explain the effect of ultrasound on Cl removal. Finally, to investigate the influence of multi-frequency ultrasound on the Cl removal performance and fly ash structure modification for various applications in environmental and material technologies, experiments were also conducted at a low ultrasonic frequency of 20 kHz with more violent collapses of cavitation bubbles.

## 2. Results and Discussion

### 2.1. Physical and Chemical Properties of Raw MSWI Fly Ash

The physical and chemical properties of raw MSWI fly ash were presented in Table 1. It was reported as a highly alkaline fly ash with pH 12.02 and a low surface area of 13.05 m^2^/g. The moisture content was 4.02%. Moreover, the Cl contamination level was extremely high, which was 32.15%, possibly from the plastic trash and food waste. Other major elements found in fly ash were 24.72% of Ca, 0.87% of Zn, 0.27% of Fe, and 0.21% of Pb. The minor chemical compositions included Cu, Ba, Cr, Cd, and Ni.

### 2.2. Results of Taguchi DOE Methodology

The results obtained from Taguchi experiments are transformed and reported as signal-to-noise (*S*/*N*) ratio using Minitab statistical software. Taguchi proposed using the *S*/*N* ratio to measure the deviation of quality characteristics from the desired values. This study aimed to optimize the Cl removal efficiency from fly ash. Therefore, the *S*/*N* ratio analysis of “larger is better” was selected, which was given in Equation (1) [33,34].
(1)$$\frac{S}{N}=-10log\frac{1}{n}\sum_{i=1}^{n}\frac{1}{y_{i}^{2}}$$
where *n* represents the number of experiments and *y* denotes the measured data.

The ideal level of the controllable factors was the level that corresponded to the largest *S*/*N* ratio, since “larger is better” was the desired scenario in Cl removal efficiency [35]. As shown in Figure 1, Figure 2 and Figure 3, the optimum conditions for Cl removal were identical for all three cases of citric acid, ascorbic acid, and EDTA-Na_2_, which were at level 4 for solution concentration, level 1 for irradiation time, level 4 for S/L ratio, and level 2 for vertical position. This indicates that the highest acid content, shortest irradiation time within 5 min, greatest S/L ratio, and the sample position of 20 mm from the bottom of the ultrasonic bath provided the favorable condition for eliminating Cl from fly ash in this Taguchi experiment. Usually, a confirmation experiment is necessary to evaluate the individual *S*/*N* ratios for the solution concentration, ultrasonic irradiation time, S/L ratio, and vertical position. However, one of Taguchi experiments contains the exact parameters. The Cl content of treated fly ash was 0.381, 0.418, and 0.410% for citric acid, ascorbic acid, and EDTA-Na_2_. With these findings, it is evident that the Cl removal in all three types of organic acid with a total reaction duration of 10 min for two phases of acid leaching and water rinsing, high acid concentration, S/L ratio of 1:10, and a sample location of 20 mm was the most efficient. The delta for each factor is the discrepancy between the highest and lowest Cl removal efficiency. The higher the delta value, the more significant the factor [32,36]. As seen from Table 2, the delta values of S/L ratio were all significantly high compared with the three other factors for all of the organic acid studies, which were 22.940, 19.585, and 13.561 for citric acid, ascorbic acid, and EDTA-Na_2_, respectively. Therefore, among the studied factors, it was discovered that the S/L ratio has the highest impact on Cl removal from fly ash.

### 2.3. ANOVA Analysis of Variance

To determine the contribution of each input parameter on the response of the process, the ANOVA analysis was performed. The ANOVA results are displayed in Appendix A for citric acid, ascorbic acid, and EDTA-Na_2_, respectively, and the response of ANOVA analysis is Cl removal efficiency. Data showed that the contribution of S/L ratio was about 90%. The other three parameters of acid concentration, ultrasonic irradiation time, and vertical position contributed to the remaining 10%. In other words, the S/L value indicating the amount of required water was the most effective factor on Cl removal by the UAW method. On the other hand, the extra amount of water used in the Cl removal process will necessitate a subsequent treatment for wastewater pollution. Therefore, to optimize the cost and effect, multiple parameters must be controlled synergistically. In this study, four controllable factors had shown the most Cl removal efficiency at the condition of high organic content, short ultrasonic irradiation time of 5 min, high S/L ratio of 1:10, and vertical position of 20 mm for all citric acid, ascorbic acid, and EDTA-Na_2_.

### 2.4. Effect of Experimental Parameters on Cl Removal

#### 2.4.1. Effect of Acid Concentration (Factor A)

In all of the three experiments of citric acid, ascorbic acid, and EDTA-Na_2_, the high acid content resulted in a slight improvement in Cl removal efficiency, as evidenced by the Taguchi DOE data, in which factor A obtained the high *S*/*N* ratio at level 4, as seen in Figure 1, Figure 2 and Figure 3a. Acid demolishes the internal structure of fly ash to a certain extent, leading to the release of Cl into the solvent [21]. Fly ash has a high concentration of metal oxides, which react with H^+^ to produce metal ions and water molecules. In addition, organic acid anions dissolve chloride salts or induce Cl^−^ displacement in fly ash [37]. Moreover, from the ANOVA analysis, the acid content contribution was around 2–3%. Therefore, this increment was not significant. The initial fly ash was strongly alkaline with a pH of 12.02. After the UAW treatment by citric acid, ascorbic acid, and EDTA-Na_2_, the pH of fly ash was 4.32, 7.45, and 9.49, respectively. It was reported that at pH < 7, the low pH resulted in a high rate of Cl elimination. There was no noticeable difference in the dechlorination rate between pH 7 and 11 [38]. As a result, for citric acid, the increase in concentration might lead to a slight increase in Cl elimination rate compared with ascorbic acid and EDTA-Na_2_. However, in all of the three case studies, the Cl removal effectiveness was about 98.8%. It could be concluded that the solution pH is not a governing factor in the Cl removal from fly ash.

#### 2.4.2. Effect of Ultrasonic Irradiation Time (Factor B)

The ultrasonic power dispersed to the ultrasound bath was 250 W, as determined by calorimetry. Cl was successfully removed after 5 min for each treatment step in this experiment. There was no significant increase in Cl dechlorination efficiency with the further increment in ultrasonic irradiation time. Small bubbles are produced, oscillated, and eventually violently collapsed as high-intensity ultrasound is irradiated into the solution. This phenomenon is known as ultrasonic cavitation and forms high fluid velocity, temperature, and pressure in the liquid, subsequently generating the mechanical and chemical effects of ultrasound [23,39]. Therefore, in this study, in addition to the Cl releasing mechanism governed by acid addition, ultrasound contributed to the leaching performance by the bubble collapse with high-velocity microjets and shockwaves, resulting in the surface excoriation, pore formation, and microchannel generation. As a result, the surface was cleaned, and the acid was able to easily permeate into fly ash particles, enhancing the Cl removal process. Furthermore, ^•^OH radicals are generated under ultrasonic irradiation from water pyrolysis at 5000 K and 1000 atm. Those active species might attach the fly ash particles or participate in chemical reactions, consequently promoting the reduction of Cl content from fly ash [40].

#### 2.4.3. Effect of S/L Ratio (Factor C)

The effect of S/L ratio on fly ash dechlorination was investigated at 1:2.5, 1:5, 1:7.5, and 1:10 for each treatment step. It was found from the Taguchi DOE experiment that the Cl removal efficiency attained maximum values at the S/L ratio of 1:10. This ratio was lower than the recommended ratio of 1:30 for heavy metal leaching by hydrochloric acid [41]. At a later time, Wang et al. used different organic acids, including citric, acetic, lactic, propionic, and butyric acid for fly ash dechlorination [38]. S/L ratio has been considered as one of the main factors in Cl leaching from fly ash. The S/L ratio ranged between 1:1 and 1:20 and the results indicated that 1:10 was the optimal ratio for Cl leaching. The other experimental conditions were acid concentration of 0.467 mol/L and 120 min of shaking. The results indicated that the final Cl content in fly ash ranged from 1.1 to 2.9%, with citric acid possessing the greatest capacity for Cl removal.

Moreover, the S/L ratio of 1:10 was applied for Cl leaching from fly ash by ultrapure water, MSW leachate, and hydrolysate of the MSW leachate with one-step and three-step leaching approaches [37]. In the one-step leaching experiment, 15 g of fly ash was added to 150 mL of solution and agitated for 3 h. In the three-step procedure, 15 g of fly ash and 50 mL of solution were shaken for 1 h and then repeated three times with 50 mL of fresh solution each time. The Cl removal efficiency of ultrapure water was 68.7 and 82.7% for the one-step and three-step leaching approach. When the MSW leachate and hydrolysate of MSW leachate were used, the Cl removal rate increased to 89.7 and 94.3% for the three-step leaching procedure. In comparison with ultrapure water, organic acids of the MSW leachate and hydrolysate of MSW leachate, including lactic, formic, acetic, propionic, isobutyric, butyric, and isovaleric acids played a significant role in the enhancement of Cl leaching efficiency. Wang et al. conducted a three-step treatment of water–lactic acid fermentation broth with S/L ratio of 1:10 and leaching time of 1 h for each step for Cl removal [14]. Following a total of 3 h treatment, the remaining fly ash content was 0.6%, which was about 1.5 times higher than this study. To date, the effect of S/L ratio on fly ash Cl removal has been studied. However, for the first time, we proved that the S/L ratio is a key determinant in Cl leaching from fly ash, which contributes more than 90% as reported by the ANOVA analysis study.

#### 2.4.4. Effect of Vertical Position (Factor D)

To investigate the effect of vertical location on the Cl elimination, the samples were placed at four positions of 5, 20, 30, and 40 mm from the bottom of the ultrasonic bath. The results indicated that at 20 mm the Cl removal efficiency was the highest among those positions, according to Taguchi experimental data. To explain the effect of sample location on the effectiveness of Cl removal, the erosion of aluminum foil in the ultrasonic bath was conducted. The white component in Figure 4 is the pit of aluminum foil with 60 μm of thickness by 5 min of ultrasonic irradiation. At 15–18 mm distance from the ultrasonic bath bottom, there were two sites of aluminum foil pitting at the centers of two nearby transducers. The ultrasonic frequency is 165 kHz and the wavelength is 9.09 mm. It was reported by Mettin that at the high-pressure locations of pressure antinodes in the ultrasonic reactor, bubbles are repelled and shifted to the positions between the pressure node and antinode due to the primary Bjerknes force [42]. As a result, at those positions, the reactive fields are comparatively high. The two erosion positions shown in Figure 4 were located at the high reactive fields between the pressure node and antinode. This explained why the obtained Cl removal efficiency was high at level 2 of 20 mm in this study.

### 2.5. Characterization of Fly Ash before and after Treatment

Figure 5 illustrates the SEM images of fly ash before and after the UAW treatment with citric acid, ascorbic acid, and EDTA-Na_2_. In Figure 5a, fly ash was hard and unclear layers covered the fly ash surface, resulting in low permeability, porosity, and surface area. Of note, after the UAW treatment, the fly ash surface was cracked and became clean and porous. Thereafter, the pores and microchannels were developed as shown in Figure 5b–d. There were two activation mechanisms for the dechlorination, including the mechanical activation of ultrasonic shockwaves, high-velocity microjets, and active radicals. In addition, the chemical activation of organic acid and ultrasonic cavitation generated active species. The collapse of ultrasonic cavitation and microbubbles unblocked the micropores and opened new pores on the surface and interlayers of fly ash, as a result of the ultrasonic penetration into the fly ash structure [43]. It favored the invasion of acid into fly ash, enlarged the pores, and thereby simultaneously increased the Cl removal efficiency.

SEM/EDS images of raw fly ash and UAW treated fly ash with different organic acids are shown in Figure 6. Ca, C, O, and Cl were the main components of raw fly ash, as can be seen in Figure 6a. Table 3 illustrates the element content of raw fly ash and treated fly ashes by different organic acids. From the map sum spectrum, it was noticed that O, Ca, Cl, and C were responsible for 99.4 wt% of the raw fly ash with 32.8, 31.7, 21.0, and 13.9 wt%, respectively. Following the two-step treatment with organic acid and water with the assistance of ultrasound, the Cl content was reduced discernibly, as demonstrated in Figure 6b–d. Ca, C, and O were identified as the major elements of UAW treated fly ash. Moreover, the element content of Cl was reported as reduced, as seen in Table 3 after the treatment by all of the applied organic acids. Therefore, the SEM and EDS results confirmed that the UAW process was successful in removing Cl from fly ash.

The XRD patterns of the raw fly ash and treated fly ashes obtained by different organic acids are illustrated in Figure 7. The high content of Cl in fly ash leads to the formation of HCl. In Taiwan, during the incineration process, semi-dry lime and activated carbon are used to treat fly ash to meet emission standards. As can be seen by the XRD patterns, there are numerous peaks of CaClOH with high intensity. It can be observed that those peaks of CaClOH of fly ashes were found to have disappeared after the treatment, indicating that CaClOH was released due to the effect of ultrasound. In addition, other peaks of NaCl, KCl, and CaSO_4_ of fly ashes were removed after the UAW process. It was in line with the map sum spectrum data of SEM/EDS and with previous XRD data [17,38]. With the dechlorination, the Ca and Cl contents in the treated fly ash became lower compared with the raw fly ash. The peak intensity of carbon significantly increased in all of the three cases of treatment by citric acid, ascorbic acid, and EDTA-Na_2_, indicating the increment in the carbon content of fly ash. This tendency was identical with the map sum spectrum data.

The Cl after leaching from fly ash enters the liquid phase, which requires further wastewater treatment procedures to meet environmental standards prior to discharge into the environment. Of note, the leachate was crystallized as the leachate remained idle overnight, resulting in the separation of Cl from the solution. In turn, this will lead to the reduction of wastewater treatment cost and time requirement, as shown in Appendix A.

### 2.6. Cl Removal Mechanism

In this study, the experimental data have shown that citric acid, ascorbic acid, and EDTA-Na_2_ all resulted in 98.8% Cl removal efficiency with a remaining Cl concentration of about 0.4%. However, the EDTA-Na_2_ concentration was utilized at 0.25 mol L^−1^, which was three times lower than the citric acid and ascorbic acid concentrations. Moreover, there is a concern that the yellowing effect on wastewater during the Cl removal process could appear by the citric and ascorbic acid due to the oxidation reactions, which might impair the wastewater quality or need further treatment. Therefore, EDTA-Na_2_ is recommended for use in real applications.

To study the mechanism of the Cl elimination process, various control experiments were performed as illustrated in Figure 8. Following a two-step water leaching process for 10 min without ultrasound and acid, the remaining Cl content in fly ash was 1.96%, which was 4.78 times higher than the Cl content obtained from the process of UAW treatment. In the system of ultrasonic treatment without organic acid, the remaining Cl content was obtained at 1.42%, which was 3.46 times higher than the UAW process. It was attributed to the strong mechanical disturbance from ultrasonic irradiation, which enables an uniform dispersion of fly ash particles into the solution, finally enhancing their collision and contact. Moreover, the free active radicals were generated, causing the mechanical and sonochemical effects of ultrasound in the reactor. These are the reasons for the enhancement of the Cl removal rate compared with the first experiment of two water leaching steps without the ultrasound. As confirmed by the XRD patterns, numerous CaClOH peaks were removed by the attack of ultrasonic waves (Figure 7). In the system using EDTA-Na_2_ leaching without ultrasound, 1.28% of Cl remained after the treatment. To investigate the mechanical and chemical effects of ultrasound on Cl removal deeply, tBuOH as a sacrificial reagent for ^•^OH was added into the solution [44]. The results indicated that the Cl removal effectiveness significantly decreased when tBuOH was introduced to the irradiated solution. The Cl content of fly ash was 0.41 and 1.17%, respectively, without and with tBuOH. The Cl removal effectiveness decreased considerably when tBuOH was introduced to the solution, demonstrating that ^•^OH radicals have a vital effect on the Cl leaching process. Therefore, both the mechanical effect of shear forces and microjet and active radicals have contributed to the Cl removal from fly ash. The map sum spectrum data also demonstrated that when adding tBuOH the Cl remaining content was 0.9%, which was higher than the Cl content of the fly ash treated by the UAW system without tBuOH. The system of both ultrasound and organic acid indicated superior Cl elimination efficiency within a short ultrasonic irradiation time. Therefore, the acid strength and the mechanical and chemical effects of ultrasound have a vital effect on Cl removal from fly ash, owing to the generated shockwaves, shear force, microjets, and ^•^OH radicals. In general, the UAW approach in this study significantly reduced the leaching time, consequently the energy demand, and treatment cost.

Briefly, a total leaching time of 10 min was required for this UAW procedure. However, it took 2 h for shaking with organic acids [38], 3 h for leaching with hydrolysate of the municipal solid waste leachate [37], and 3 h for leaching with lactic acid fermentation broth [14]. The as-developed leaching method also revealed a superior capability in Cl removal compared with the high-cost method of thermal treatment [17], which consumed 1 h for roasting at 1050 °C to eliminate Cl from 8.6 to 1.47%. The water-washing procedure was also performed. Moreover, 1.81 and 0.60% Cl content were obtained after 1.5 h of water-flushing at room temperature and 90 °C, respectively. Aeration and carbonation methods were combined with the water leaching method, resulting in the increase in Cl removal efficiency and leading to the decrease in Cl content from 8.6 to 0.43% and 0.26%, respectively. All of these procedures required extra energy and consumed a large amount of time. Therefore, the synergistic effect of ultrasound and organic acid provided a significant pathway for Cl elimination from fly ash. The UAW approach might be considered as a cost-effective solution for Cl removal using inexpensive RO water, short reaction time, and less energy demand.

### 2.7. Synergistic Effect of Multi-Frequency Ultrasound

The ultrasonic frequency is a predominant influence factor on the mechanical and chemical effects of ultrasonic cavitation. At low frequencies, it is known that the cavitating bubbles are large and severely collapse, generating a powerful jet flow, shock waves, and shearing stress, which easily lead to the strong physical activation of morphology and surface of materials in an ultrasonic reactor. On the other hand, Koda et al. reported that at an ultrasonic frequency in the range of several hundred kHz, the number of collapsing bubbles is high and the sonochemical reaction is strong, whereas under 100 kHz the sonochemical effect is unaffected by the frequency [45]. Both the ultrasonic mechanical and chemical effects highly contributed to the Cl removal in the UAW process. To investigate the effect of ultrasonic frequency on the fly ash dechlorination, a 20 kHz ultrasound with the ultrasonic power of 250 W was used in this study, along with the same experimental conditions at 165 kHz. Cl was eliminated from 32 to 0.36% by the UAW process at 20 kHz. In Figure 9a, the SEM image of treated fly ash showed a clean surface, dense micropores, and a porous structure. Compared with Figure 5, the fly ash surface was even more clear, and the fly ash structure was more porous with denser micropores and microchannels.

In Figure 9b, it can be seen that there was a discernible change in Cl content, as compared with Figure 5a of raw fly ash. This indicates that Cl was massively diminished from fly ash. At 20 kHz, the highest contributed elements were Ca (19.8%), C (20.5%), and O (49.3%), which were the same tendency in the raw fly ash and UAW treated fly ash at 165 kHz. The Cl content of raw fly ash was 21 wt% and reduced to 0.6 and 0.5 wt% at 165 and 20 kHz, respectively. This pattern corresponded to the Cl removal efficiency. Table 4 showed the specific surface area of raw and different treated fly ashes. The surface area of raw fly ash was 13.05 m^2^/g. The surface area increased to 91.47 and 98.06 m^2^/g after UAW treatment at 165 and 20 kHz with a 5 min treatment time for each step, respectively, indicating that both low and high ultrasonic frequencies were able to enhance the surface morphology, as well as remove the contaminants of fly ash for further applications. As the ultrasonic irradiation time increases to 20 min for each treatment step, the surface area decreases to 27.02 and 27.59 m^2^/g at 165 and 20 kHz. With the extended irradiation time, extra ultrasonic vibration allowed for the separation of fly ash particles. Then, these fragments aggregated and blocked the internal pores of fly ash, which consequently reduced the fly ash surface area [46]. Moreover, this explained the reason that the optimal ultrasonic irradiation time for Cl removal was at 5 min (Part 2.4.2) and Cl removal efficiency did not enhance as the treatment time further increased. Therefore, in reality, an ultrasonic reactor with multi-frequency transducers should be constructed and the ultrasonic irradiation time needs to be well controlled for the UAW treatment to achieve a high Cl removal performance and an efficient material modification to get a high-quality material for other applications in environmental remediation and material applications.

## 3. Materials and Methods

### 3.1. Materials

Citric acid (99.5–100.5%) and ethylenediaminetetraacetic acid disodium salt dihydrate (EDTA-Na_2_—97%) were purchased from Sigma-Aldrich, MO, USA. Ascorbic acid (99%) and *tert*-butyl alcohol (tBuOH—99%) were supplied by Thermo Fisher Scientific (Waltham, MA, USA). All of the reagents were of analytical grade and used without any further purification. The fly ash used in this study was collected from MSWI plant in Taiwan. RO water was used to prepare all of the aqueous solutions.

### 3.2. Experiment Procedure

To reduce the cost for the real application of this Cl leaching method, RO water was employed for the whole study. RO water was saturated with air prior to use in the experiments. There were two steps for Cl removal, including organic acid leaching and water rinsing assisted by ultrasound. At the organic acid leaching step, 50 mL of solution at a certain concentration and a specific amount of fly ash based on the solid/ liquid ratio were placed in a 250 mL glass beaker and irradiated by a combination of ultrasonic wave and stirring. After sonication, the fly ash and liquid were separated by a vacuum filtration pump. The water rinsing stage followed the same procedure using the as-irradiated fly ash from the first leaching step. However, rather than organic acid, we employed air-saturated RO water to further remove Cl, other contaminants, as well as residual organic acid from the fly ash. After the leaching and rinsing procedures, the fly ash was dried in an oven at 80 °C for 2 h, cooled down to ambient temperature, ground into a fine powder, and stored in a bag for further analysis.

### 3.3. Design of Experiment by Taguchi Methodology

The effectiveness of Cl removal from fly ash in the ultrasonic bath is influenced by a variety of parameters and interactions. The Taguchi methodology was applied to find the most appropriate and optimum parameter combinations with the minimum number of trials. In this study, the impact of four major controllable variables was used, including acid concentration (factor A), ultrasonic irradiation time (factor B), fly ash/solution (S/L) ratio (factor C), and vertical position of sample from the bottom of the ultrasonic bath (factor D) with four levels for each factor. The liquid height in the ultrasonic bath was controlled at 50 mm. The controllable factors and their levels are illustrated in Table 5. These levels were decided according to previous researches [38,40]. An L-16 orthogonal array was designed for the study, as shown in Table 6. A total of 16 experimental tests for one run were performed for each organic acid, with each experiment duplicated to avoid experimental errors.

### 3.4. Calorimetry

Calorimetry was used to measure the ultrasonic power ($P_{U}$) dispersed into the solution in the ultrasonic bath [47]. A thermal couple was placed into the ultrasonic bath to monitor the temperature increase. Calorimetry was carried out at least 8 times. The increase in the temperature of the solution was used to estimate the power delivered into the solution.
(2)$$P_{U}\left(W\right)=\frac{dT}{dt}C_{P}M$$
where $\frac{dT}{dt}$ (K/s) denotes the temperature rise per sonication time, $C_{P}$ is the specific heat capacity of solvent at constant pressure and 25 °C ($C_{P}=4.1853\;J\cdot g^{-1}\cdot K^{-1}$), and *M* represents the mass solvent (g).

### 3.5. Analytical Methods

The moisture content was measured by the weight difference of fly ash before and after oven drying at 105 °C for 24 h. The element contents of fly ash before and after the treatment were determined by X-ray fluorescence (XRF, Olympus NDT, Waltham, MA, USA). The surface morphology was characterized by field-emission scanning electron microscopy (FE-SEM, JSM-7610FPlus, JEOL, Tokyo, Japan) with an acceleration voltage of 15 kV. The distribution of elements was examined by energy dispersive X-ray (EDS) spectrometer (Ultim Max TEM, JEOL). The specific surface area (S_BET_) was determined by nitrogen gas adsorption in the *P*/*P*_0_ range of 0.05–0.2 using the Brunauer–Emmett–Teller (BET) equation. The phase structure of fly ash before and after the treatment was determined using a Bruker D825A advance X-ray diffractometer (XRD) with Cu Ka radiation (λ = 1.5405 Å) at a voltage and current density of 40 kV and 40 mA. The scanning step was 0.05° and the scanning speed was 2°/min in the scanning range of 10–80°.

## 4. Conclusions

The ultrasonic-assisted organic acid treatment was implemented to remove Cl from MSWI fly ash to facilitate the reuse and recycling of fly ash for material production in the construction industry and environmental applications. The Taguchi methodology and ANOVA analysis were used to study the effect of organic acid concentration, ultrasonic irradiation time, solid/liquid ratio, and vertical position of fly ash in the ultrasonic bath on Cl removal performance. The research indicated that the solid/liquid ratio was the most important factor, which contributed to more than 90% of dechlorination. The UAW presented as a simple procedure with less energy demand due to a short reaction time of 5 min for Cl elimination compared with the other conventional processes. Physical characterization showed the clean and porous material after the treatment, indicating that it could be advantageous in various materials and environmental applications. In a real application, a multi-frequency ultrasound is recommended for fly ash removal by this UAW process, owing to the strong physical activation and high sonochemical effect at low and high frequency, respectively.

## Figures and Tables

**Figure 1 molecules-27-02289-f001:**
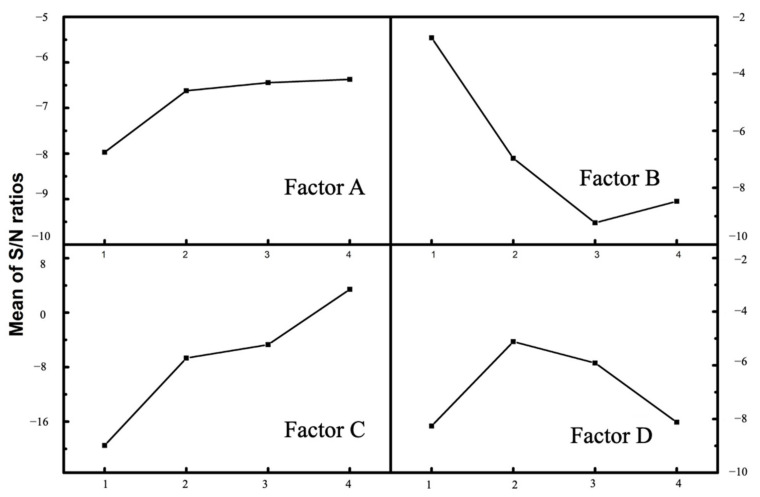
*S*/*N* ratio plots for the effects of experimental parameters for citric acid.

**Figure 2 molecules-27-02289-f002:**
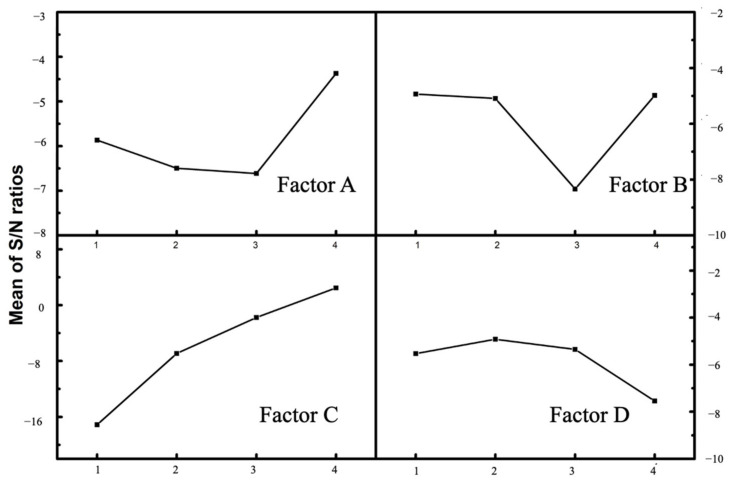
*S*/*N* ratio plots for the effects of experimental parameters for ascorbic acid.

**Figure 3 molecules-27-02289-f003:**
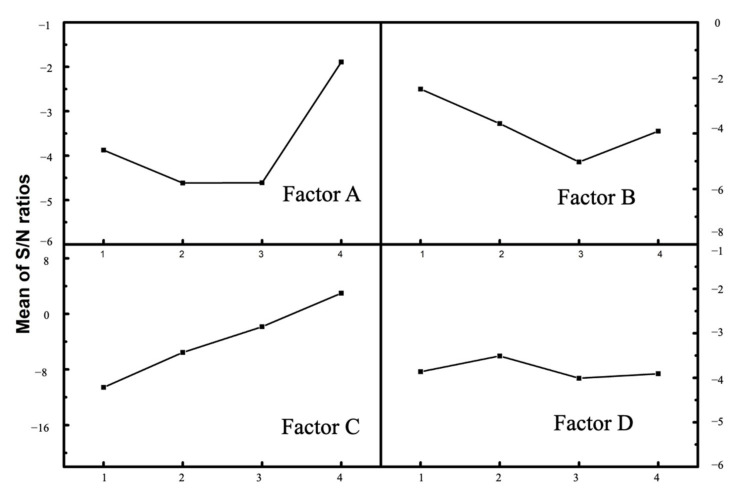
*S*/*N* ratio plots for the effects of experimental parameters for EDTA-Na_2_.

**Figure 4 molecules-27-02289-f004:**
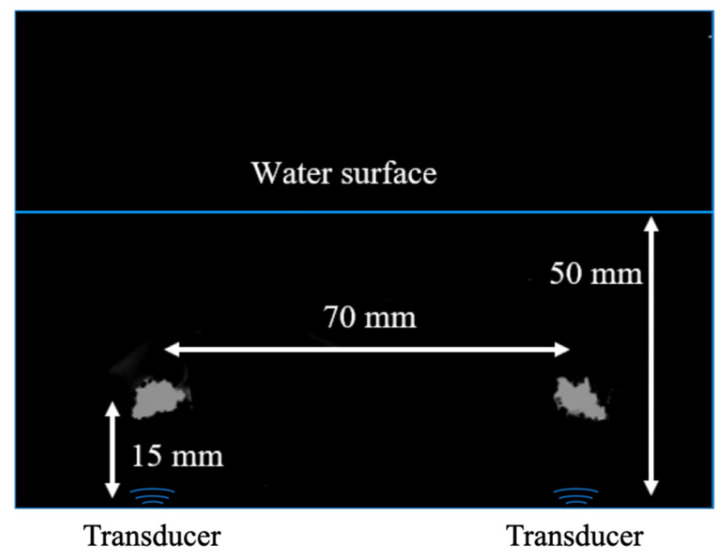
Aluminum foil erosion at 165 kHz for 5 min.

**Figure 5 molecules-27-02289-f005:**
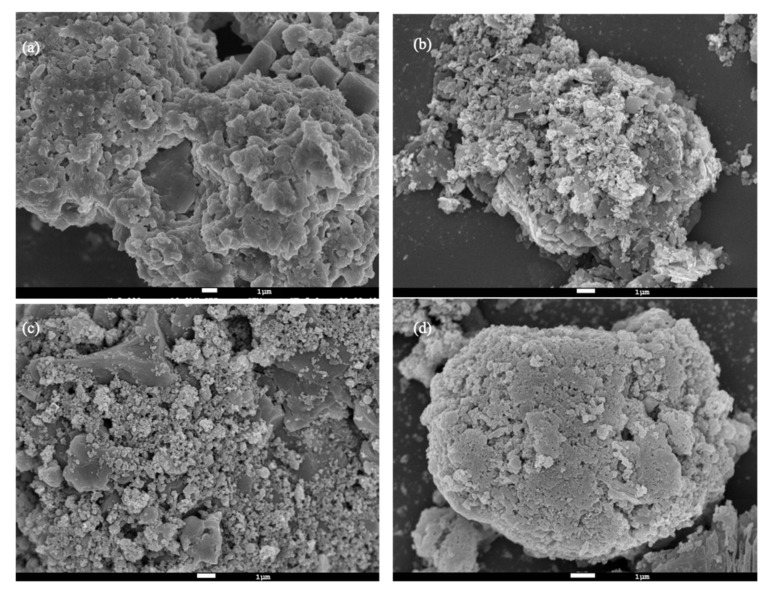
SEM image of raw fly ash (**a**), UAW by citric acid treated fly ash (**b**), UAW by ascorbic acid treated fly ash (**c**), and UAW by EDTA-Na_2_ treated fly ash (**d**).

**Figure 6 molecules-27-02289-f006:**
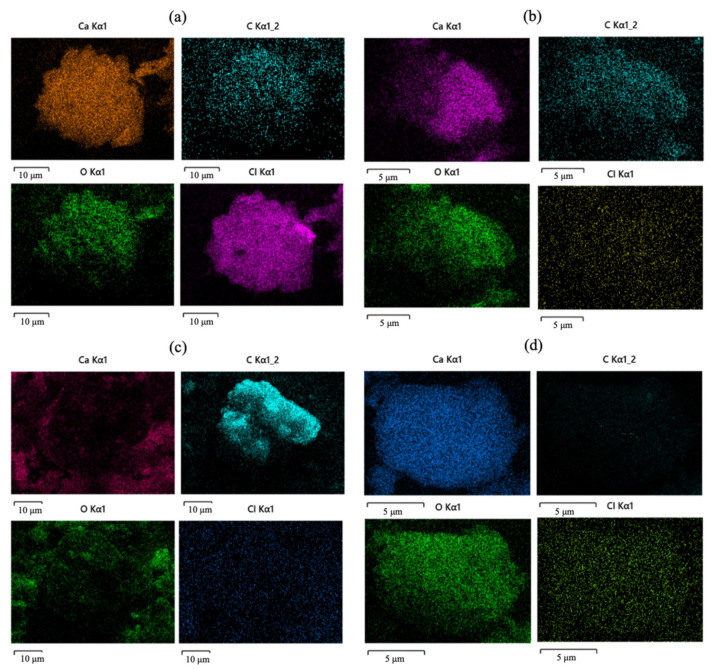
SEM/EDS of raw fly ash (**a**), UAW by citric acid treated fly ash (**b**), UAW by ascorbic acid treated fly ash (**c**), and UAW by EDTA-Na_2_ treated fly ash (**d**).

**Figure 7 molecules-27-02289-f007:**
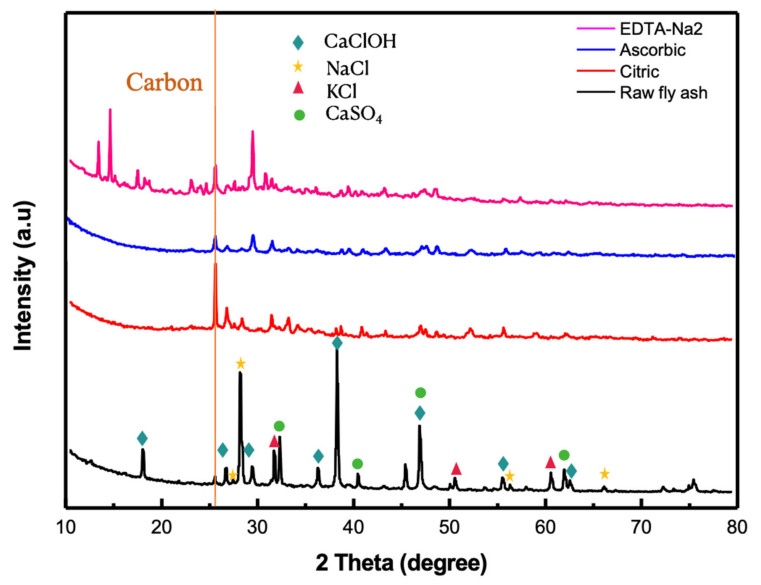
XRD patterns of fly ashes.

**Figure 8 molecules-27-02289-f008:**
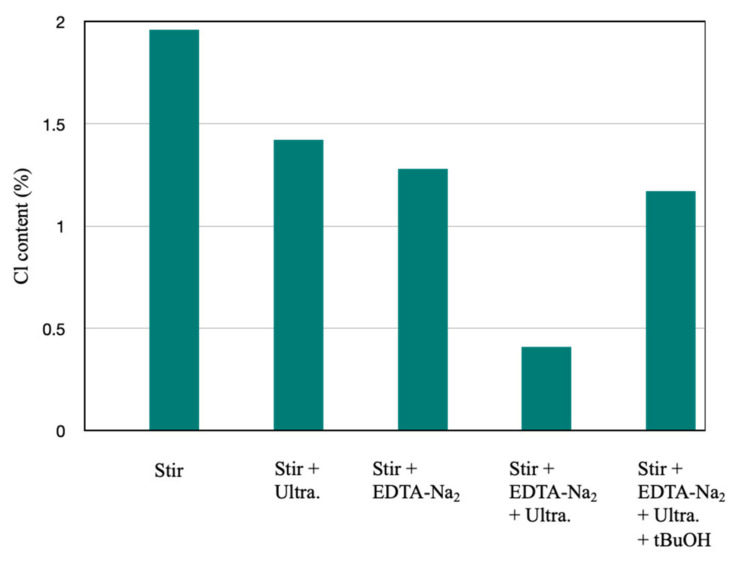
Cl contents after different treatment conditions.

**Figure 9 molecules-27-02289-f009:**
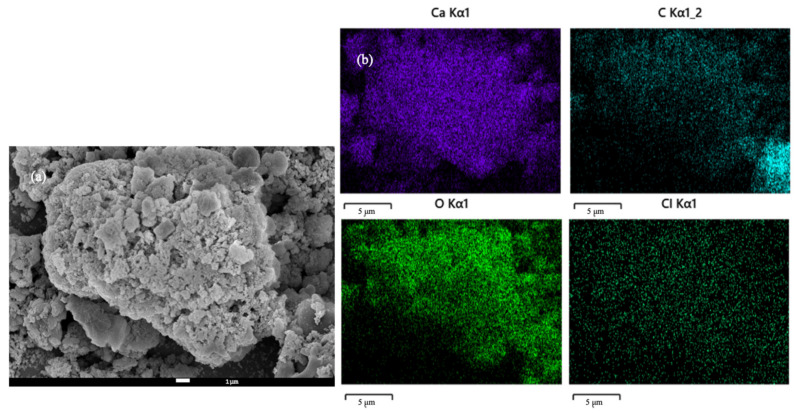
SEM image (**a**) and SEM/EDS (**b**) of UAW treated fly ash using EDTA-Na_2_ at 20 kHz.

**Table 1 molecules-27-02289-t001:** Physical and chemical properties of raw MSWI fly ash.

Physical Properties			Chemical Properties									
Moisture content (%)	pH	Surface area (m^2^/g)	Ca (%)	Cl (%)	Zn (%)	Fe (%)	Pb (%)	Cu (%)	Ba (%)	Cr (%)	Cd (%)	Ni (%)
4.02	12.02	13.05	24.72	32.15	0.87	0.27	0.21	0.075	0.009	0.011	0.015	0.007

**Table 2 molecules-27-02289-t002:** Delta values from Taguchi analysis.

	Delta			
	A	B	C	D
Citric	1.599	6.498	22.940	3.144
Ascorbic	2.239	3.402	19.585	2.625
EDTA-Na_2_	2.724	2.630	13.561	0.501

**Table 3 molecules-27-02289-t003:** Element content of fly ashes.

	Wt%			
	O	Ca	Cl	C
Raw fly ash	32.8	31.7	21	13.9
Citric	49.7	15.6	0.3	26
Ascorbic	31.8	10.3	0.3	44.7
EDTA-Na_2_	49.5	20.3	0.3	22.6

**Table 4 molecules-27-02289-t004:** Surface area of fly ashes.

	Surface Area (m^2^/g)
Raw fly ash	13.05
EDTA-Na_2_ at 165 kHz, 250 W, 5 min	91.47
EDTA-Na_2_ at 165 kHz, 250 W, 20 min	27.02
EDTA-Na_2_ at 20 kHz, 250 W, 5 min	98.06
EDTA-Na_2_ at 20 kHz, 250 W, 20 min	27.59

**Table 5 molecules-27-02289-t005:** Factors and levels used in the experiments.

Symbol	Factors	Unit	Levels				
				1	2	3	4
A	Conc.	mol L^−1^	Citric	0.1	0.25	0.5	0.75
			Ascorbic	0.1	0.25	0.5	0.75
			EDTA-Na_2_	0.01	0.05	0.1	0.25
B	Time	min		5	10	15	20
C	S/L ratio	-		1:2.5	1:5	1:7.5	1:10
D	Position	mm		5	20	30	40

**Table 6 molecules-27-02289-t006:** L16 orthogonal array for Cl removal by UAW.

Exp. No.	Factors			
	A	B	C	D
1	1	1	1	1
2	1	2	2	2
3	1	3	3	3
4	1	4	4	4
5	2	1	2	3
6	2	2	1	4
7	2	3	4	1
8	2	4	3	2
9	3	1	3	4
10	3	2	4	3
11	3	3	1	2
12	3	4	2	1
13	4	1	4	2
14	4	2	3	1
15	4	3	2	4
16	4	4	1	3

## Data Availability

All data are available in the manuscript and from the authors.

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
