# Peer review of "Sustainable Recovery of Valuable Nanoporous Materials from High-Chlorine MSWI Fly Ash by Ultrasound with Organic Acids"

_molecules, 2022, doi:10.3390/molecules27072289_

Round 1
Reviewer 1 Report
In this manuscript, authors used a acid with assisted ultrasound to eliminate Cl from fly ash.The method has the engineering value, and it could help improve the Municipal solid waste. But, there was some problems to be solved.
1 the ultrasound power should be given in paper, the power was very important for the removal efficiency of Cl.
2 in section 2.4.3, it mentioned the ratio were 1:2.5, 1:5, 1:7.5, and 1:10, but the good result was 1:10, the result of 1:15 or 1:20 need to add, or else the ‘The S/L ratios ranged between 1:1 and 1:20 and the results indicated that 1:10 was the optimal ratio for Cl leaching’ was not supported by the manuscript investigation.
3 in fig.7, the noise was obvious, it was suggested data denoising.
4 in table.4, the ultrasound power should be given, it was important data. If power was different, the results may be varied.
Author Response
Responses to Editor and Reviewer’s Comments
Ms. No. molecules-1651506
Title: Sustainable recovery of valuable nanoporous materials from high-chlorine MSWI fly ash by ultrasound with organic acids
Dear Editor and reviewer,
We appreciate you for your precious time in reviewing our paper and providing valuable comments. It was your valuable and insightful comments that led to possible improvements in the current version. The authors have carefully considered the comments and tried our best to address every one of them. We hope the manuscript after careful revisions meet your high standards. The authors welcome further constructive comments if any.
Below we provide the point-by-point responses. All modifications in the manuscript have been highlighted in red.
Sincerely,
Tam Thanh Nguyen, Ph.D.

Reviewer 2 Report
1) Title: Sustainable recovery of valuable nanoporous materials from high-chlorine MSWI fly ash by ultrasound with organic acids
Comment: Please include the statements about "valuable nanoporous" in the manuscript.
2) Line 136. surface area provided in the text does not match with the one you mentioned in Table 1.
3) please cite the paper from where you received the information that you provided in Table 1
4) Line 163. Please mention in the text what delta values stands for
5) Line 167. Why did you cite a reference there [37].
6) Table 2 What are those A, B, C, and D
7) Line 223 Please include a reference
8) Figure 5. why the scale is different for the raw fly ash?
9) Figure 6. Why the scales are different sometimes 5 or 10
10) Line 423. Delete "analytical methods"
11) section 3.4 You need elaborate on this part. Please provide more information
12) On what basis do you choose those ranges (levels) for each factor in table 5. if you chose those ranges from some studies, please cite those papers.
Author Response
Responses to Editor and Reviewer’s Comments
Ms. No. molecules-1651506
Title: Sustainable recovery of valuable nanoporous materials from high-chlorine MSWI fly ash by ultrasound with organic acids
Dear Editor and Reviewer,
We appreciate you for your precious time in reviewing our paper and providing valuable comments. It was your valuable and insightful comments that led to possible improvements in the current version. The authors have carefully considered the comments and tried our best to address every one of them. We hope the manuscript after careful revisions meet your high standards. The authors welcome further constructive comments if any.
Below we provide the point-by-point responses. All modifications in the manuscript have been highlighted in red.
Sincerely,
Tam Thanh Nguyen, Ph.D.

Round 2
Reviewer 2 Report
The responses and modifications made to the MS are acceptable, and I suggest that this manuscript be approved for publication in the Molecules journal (ISSN 1420-3049). Thank you